# Community volunteer participation and its determinants during respiratory infectious disease outbreaks in China: A cross-sectional study across multiple provinces

Huizi Jin[1☯], Xueting Ding[2☯], Zhijing Li[1], Juan Cao[3], Xinying Sun[1], Ying Ji[1]*, Fang Ning[4]*

**1** School of Public Health, Peking University, Beijing, People's Republic of China, **2** Department of Health, Society, & Behavior, Joe C. Wen School of Population & Public Health, Susan and Henry Samueli College of Health Sciences, University of California, Irvine, Irvine, California, United States of America, **3** Collaborative Innovation Center of Assessment Toward Basic Education Quality, Beijing Normal University, Beijing, People's Republic of China, **4** Beijing Center for Diseases Prevention and Control, Beijing, People's Republic of China

☯ These authors contributed equally to this work.
* 0016179051@bjmu.edu.cn (YJ); cdcnf@163.com (FN)

## Abstract

### Background

Community volunteering plays a crucial role in strengthening public health emergency response, particularly during respiratory infectious disease outbreaks. However, limited research has examined the extent of resident participation and the factors influencing engagement in such efforts. This study investigated the prevalence and determinants of community volunteering during the COVID-19 pandemic and other respiratory infectious disease outbreaks in China.

### Method

We conducted a cross-sectional online survey and included data from 1,023 residents residing in five provinces in China (Beijing, Guangdong, Heilongjiang, Hubei, and Yunnan) between early 2020 and March 2023. Participants reported their socio-demographic factors, volunteer activities, related motivations and barriers during the pandemic. We used logistic regression models to identify factors associated with volunteering.

### Results

Of the respondents, 65.9% participated in community volunteering during the pandemic, with the most common roles related to nucleic acid testing. The primary motivation for volunteering was value expression. Main barriers to participation included a lack of time, limited professional skills, and concerns about the risk of

**Data availability statement:** The minimal de-identified dataset used in the current study is available from [https://figshare.com/articles/dataset/Data_for_the_study_Community_volunteer_participation_and_its_determinants_during_respiratory_infectious_disease_outbreaks_in_China_a_cross-sectional_study_across_multiple_provinces_/29602721?-file=56387153].

**Funding:** The National Social Science Found of China (Grant No. 22AZD077) supported XS and YJ with the project (http://www.nopss.gov.cn/GB/index.html). The funders didn't play any role in the study design, data collection and analysis, decision to publish, or preparation of the manuscript.

**Competing interests:** The authors have declared that no competing interests exist.

infection. Higher odds of participation were observed among respondents who held at least a bachelor's degree (AOR = 2.58, 95% CI: 1.21–5.48), worked in community (AOR = 4.32, 95% CI: 2.56–7.28) or health-care roles (AOR = 2.48, 95% CI: 1.31–4.67), were Communist Party members (AOR = 1.68, 95% CI: 1.07–2.64), or had volunteered regularly before 2020 (AOR = 3.02, 95% CI: 2.51–3.64), while single/divorced/widowed individuals had lower odds of participation (AOR = 0.60, 95% CI: 0.39–0.94) (all p < 0.05).

## Conclusion

Community volunteering could constitute a substantial auxiliary workforce during respiratory epidemics in China. Integrating volunteers into emergency preparedness may require institutionalized training programs, transparent management structures, as well as legal and policy safeguards that recognize volunteers' contributions and mitigate perceived risks. Such measures are likely to strengthen community resilience in future public-health emergencies.

## Introduction

Volunteer service, defined as voluntary, unpaid public-interest work undertaken by individuals or organized groups, has long been recognized as a mechanism for strengthening social cohesion and civic engagement [1–3]. Recent national guidelines on volunteer service development in China position volunteering as a marker of social progress and an integral component of emergency preparedness [4]. These policies, together with a steady increase in social emergency deployments reaching about 1.8 million between 2018 and 2020 [4], underscore the strategic importance of community volunteers for public-health response. During the COVID-19 pandemic, community volunteers were mobilized through formal government channels (including neighborhood committees and street offices) and informal community networks, which created a hybrid volunteer mobilization system that leveraged existing social infrastructure while expanding capacity through grassroots engagement. Although formal integration mechanisms are not yet fully established in China, the legal framework for volunteer integration in emergencies continues to evolve based on these practical experiences.

Research has shown that engagement in volunteering reflects a mixture of internal and external determinants. Internally, sociodemographic factors such as sex, age, occupation, marital status, and educational attainment, as well as individual motivations including learning, career advancement, value expression, self-enhancement, self-protection, and social connection, associate with whether and how people volunteer [5–9]. Externally, supportive family and peer norms, credible organizational structures, and clear incentive or safeguard mechanisms foster participation, whereas deficiencies in recruitment channels, training, and rights protection can dampen enthusiasm [7,10–13].

The COVID-19 outbreak in early 2020 brought these issues into sharp relief. As healthcare and community staff became overstretched [14], volunteers stepped in to provide logistical support, health education, check-point staffing, and nucleic-acid testing, illustrating the untapped capacity of China's community volunteer corps. Although China's State Council Joint Prevention and Control Mechanism announced in February 2023 that the domestic COVID-19 epidemic was largely under control [7,15], SARS-CoV-2, influenza A, and Mycoplasma pneumoniae continue to circulate, emphasizing the need for a sustainable, well-trained volunteer workforce for future respiratory epidemics.

Despite the prominent role that volunteers played in China's epidemic response, systematic evidence on who participated, in what roles, and under which conditions remain scarce. Prior studies have tended to focus on single municipalities or on general (non-emergency) volunteering, limiting regional coverage and external validity. To address these gaps, the present study surveyed residents in five provinces in China between early 2020 and March 2023. Our objectives were to (i) describe the scope, modalities, motivations, and barriers of community volunteering during consecutive respiratory infectious-disease outbreaks; (ii) identify sociodemographic and experiential factors associated with participation; and (iii) generate evidence to inform incentive schemes, training programs, and management frameworks that could integrate community volunteers more effectively into future public-health emergency preparedness efforts in China and comparable contexts.

## Materials and methods

### Study design

We conducted a cross-sectional, province-stratified online survey during March 2nd to 31st, 2023. The target population comprised adults residing in five provinces including Beijing, Guangdong, Heilongjiang, Hubei, and Yunnan. The selection of survey provinces considered multiple criteria including selecting areas with the highest cumulative COVID-19 case counts before November 2022 from each of the five major geographical regions (northern, southern, eastern/central, western, and north-eastern), economic development diversity, and geographical distribution to ensure representative coverage of mainland China [16]. We used the formula $N = [Z^2_{1-\alpha/2}P(1-P)]/d^2$, where $\alpha = 0.05$, $Z_{1-\alpha/2} = 1.96$, the acceptable margin of error $d = 0.10$, and the 30% prevalence of relevant outcomes during the COVID-19 pandemic to estimate the target sample size. As a result, we got a minimum required sample of 81 participants per province [16].

### Recruitment and sampling

We used quota sampling to approximate the national population distribution reported in the Seventh National Census in China. Quotas were set for sex (men 51%, women 49%), age (18–39 years 40%, 40–70 years 60%), and Chinese household registration (urban 65%, rural 35%). Given the post-pandemic context and associated time and resource constraints, we implemented the recruitment through the Wenjuanxing online survey platform to enhance sample representativeness and ensure timely data collection. The online platform implemented comprehensive privacy protection measures including IP address anonymization, secure server protocols, and encrypted data transmission to ensure participant confidentiality throughout the data collection process. Eligible participants had to (i) be 18–70 years of age, (ii) have resided in the selected province for at least six months, (iii) have internet access and ability to read in Chinese, (iv) reside locally during the survey period, and (v) provide electronic informed consent. After the pre-screening process, approximately 200 respondents per province were invited, and we got 1,026 initial responses.

To ensure data quality, the questionnaire included attention-check items, required a minimum completion time of 5 minutes and at least 3 seconds per item, and permitted only one submission per device. Four questionnaires were excluded for implausible completion times or logical inconsistencies, leaving 1,023 valid cases (effective response rate 99.8%). This satisfied our sample size calculation requirement of 81 participants per province (405 in total). We set the online questionnaire platform to require mandatory completion of all questions, with participants unable to submit responses without answering every item, thereby ensuring complete data collection for all study variables. More details on our data collection

procedures can be found elsewhere [16]. The study protocol was approved by the Peking University Institutional Review Board (IRB00001052–22171). Electronic written informed consent for participation and publication was obtained from all participants through the online survey platform, immediately prior to initiating the questionnaire.

## Measures

**Outcome variable.** The primary outcome was participation in community epidemic-control volunteering between January 2020 and March 2023. Respondents first indicated whether they had taken part in any such activity (0 = no, 1 = at least once). Those answering "yes" were then asked to specify the type of service and their cumulative hours of service, reported to the nearest hour. The type of service was classified into five predefined categories adapted from the framework of Mak et al. [5] and current community practice: 1) nucleic-acid testing assistance (e.g., QR-code registration, queue management, and temperature monitoring), 2) registration of residents' entry to and exit from the community (including visitor screening and resident identification verification), 3) temperature or health-code screening at community checkpoints (including symptom monitoring and health status verification), 4) delivery of supplies to quarantined households (including food, medication, and essential items), and 5) dissemination of epidemic-prevention information (including distributing educational materials, conducting health education sessions, and providing psychosocial support to community members).

**Independent variables.** Eight socio-demographic characteristics were included on the basis of previous volunteering research [7,17]: sex, age, household registration (urban/rural), marital status, educational attainment, employment category (community-street-office staff, health-care-related staff, other), political affiliation, and frequency of volunteering before 2020 (five-point Likert scale from "never" to "very often"). Income was not included as a variable due to cultural sensitivity concerns in Chinese contexts, where individuals may be reluctant to accurately report income levels. Employment category was thus included as a proxy measure to capture relevant socioeconomic status differences.

To explore factors beyond basic sociodemographic characteristics, we incorporated three supplementary domains derived from published instruments on volunteer engagement: (i) perceived barriers to volunteering [18]; (ii) volunteering motivations [9,10]; and (iii) general attitude toward volunteering. Specifically, perceived barriers were assessed with nine dichotomous ("yes/no") items adapted from Guo et al. [18], covering lack of time, insufficient professional skills, fear of infection, inadequate organizational support, family obligations, and five other commonly reported constraints. Motivations were measured with eleven multi-select items based on the Volunteer Functions Inventory and subsequent extensions [9,10]. Sample options included "helping people in need," "fulfilling social responsibility," and "promoting community civility," alongside statements related to personal growth, career development, and value expression. Overall attitude toward community volunteering was captured with a single item offering three response categories: "support," "oppose," or "undecided." All barrier and motivation items allowed multiple selections, and attitude was recorded as a categorical variable for subsequent descriptive analysis.

## Statistical analysis

Data was analyzed with SPSS, version 26.0. Descriptive statistics and multiple-response tabulations summarized volunteer roles, motivations, and perceived barriers. Binary logistic regression was used to estimate crude and adjusted odds ratios for the association between the independent variables and our outcome. Odds ratios (ORs) with 95% confidence intervals (CIs) were reported. All tests were two-sided, and statistical significance was set at α = 0.05. Prior to conducting multivariable logistic regression, we assessed multicollinearity among independent variables using variance inflation factors (VIFs). All VIF values ranged from 1.032 to 1.270 (S1 Table), well below the threshold of 5, indicating acceptable levels of multicollinearity. Model fit was assessed using Nagelkerke $R^2$. No statistical weights were applied to the data, as our quota sampling approach was designed to approximate national demographic distributions. To account for multiple comparisons, Bonferroni correction was applied with appropriate adjustment of significance levels.

## Results

### Participant characteristics

Among the 1,023 questionnaires, men accounted for 50.2% of respondents and women for 49.8%. The age distribution was evenly split across three categories: 18–34 years (32.4%), 35–44 years (33.9%), and 45–70 years (33.7%). Most participants held urban household registration (61.6%), were married or living with partner (73.2%), and had attained at least a bachelor's degree (68.0%). In terms of occupation, 250 (24.4%) were street- or community-level employees, 121 (11.8%) students, 109 (10.7%) healthcare staff. 297 of the participants (29.0%) self-identified as Communist Party members (Table 1).

### Volunteering experience during respiratory epidemics

Overall, 92.4% of respondents expressed support for community volunteering. Between January 2020 and March 2023, 674 individuals (65.9%) reported participating in at least one epidemic-control volunteer activity. The median cumulative service time among volunteers was 50 hours (IQR 20–50 hours). The most common roles were nucleic-acid testing support (78.8%), entry–exit registration for residential compounds (65.1%), and procurement or delivery of daily supplies (58.2%). About one-third (35.3%) provided psychosocial support to residents in home isolation (Table 2).

### Motivations for and barriers to volunteering

Volunteers most frequently cited helping people in need (78.2%), fulfilling social responsibility (66.2%), and promoting community civility (57.4%) as their reasons for engagement. By contrast, fewer participants selected networking

**Table 1. Sociodemographic characteristics of respondents (N = 1,023).**

| Characteristic | | n (%) | | |
| --- | --- | --- | --- | --- |
| | | **Total** | **Did not volunteer** | **Volunteered** |
| **Sex** | Men | 514 (50.2%) | 159 (30.9%) | 355 (69.1%) |
| | Women | 509 (49.8%) | 190 (37.3%) | 319 (62.7%) |
| **Age (years)** | 18-34 | 331 (32.4%) | 135 (40.8%) | 196 (59.2%) |
| | 35-44 | 347 (33.9%) | 112 (32.3%) | 235 (67.7%) |
| | 45-70 | 345 (33.7%) | 102 (29.6%) | 243 (70.4%) |
| **Household registration** | Urban | 630 (61.6%) | 201 (31.9%) | 429 (68.1%) |
| | Rural | 393 (38.4%) | 148 (37.7%) | 245 (62.3%) |
| **Marital status** | Single/ divorced/ widowed | 274 (26.8%) | 142 (51.8%) | 132 (48.2%) |
| | Married/ living with partner | 749 (73.2%) | 207 (27.6%) | 542 (72.4%) |
| **Educational attainment** | Bachelor's degree or higher | 696 (68.0%) | 216 (31.0%) | 480 (69.0%) |
| | High-school/ technical college/ associate | 275 (26.9%) | 110 (40.0%) | 165 (60.0%) |
| | Primary/ junior secondary | 52 (5.1%) | 23 (44.2%) | 29 (55.8%) |
| **Employment** | Healthcare staff | 109 (10.7%) | 16 (14.7%) | 93 (85.3%) |
| | Student | 121 (11.8%) | 63 (52.1%) | 58 (47.9%) |
| | Street/ community staff | 250 (24.4%) | 22 (8.8%) | 228 (91.2%) |
| | Other | 543 (53.1%) | 248 (45.7%) | 295 (54.3%) |
| **Political affiliation** | Communist Party member | 297 (29.0%) | 40 (13.5%) | 257 (86.5%) |
| | Democratic party member | 6 (0.6%) | 1 (16.7%) | 5 (83.3%) |
| | Communist Youth League member | 137 (13.4%) | 67 (48.9%) | 70 (51.1%) |
| | Unaffiliated | 583 (57.0%) | 241 (41.3%) | 342 (58.7%) |
| **Total** | | 1023 (100%) | 349 (34.1%) | 674 (65.9%) |

**Table 2. Volunteer activity types among participants (n = 674).**

| Activity type | Frequency | Response%[a] | Case%[b] |
|---|---|---|---|
| Nucleic-acid testing support | 531 | 31.1% | 78.8% |
| Entry–exit registration | 439 | 25.7% | 65.1% |
| Procurement/ delivery of supplies | 392 | 23.0% | 58.2% |
| Psychosocial support (home isolation) | 238 | 13.9% | 35.3% |
| Other activities | 107 | 6.3% | 15.9% |
| Total | 1707 | 100.0% | 253.3% |

[a]Percentage of all responses (multiple selections allowed): Number of times a category was chosen ÷ Total number of choices made across all categories.

[b]Percentage of volunteers selecting the item: Number of volunteers who selected the category ÷ Total number of volunteers in the study.

opportunities (26.4%), material or academic incentives (23.4%), or meeting formal Party-service requirements (26.6%) (Table 3). When the entire sample was queried about barriers, lack of time (60.3%), insufficient professional skills (49.5%), fear of infection (33.2%), and limited access to information (32.9%) emerged as the most commonly perceived obstacles, whereas distrust of organizations (8.1%), disinterest (11.1%), or simple unwillingness (6.3%) were seldom endorsed (Table 4).

## Factors independently associated with volunteering

Multivariable logistic regression indicated that five characteristics remained significant after adjustment for all covariates (Table 5). This model demonstrated good fit (Nagelkerke $R^2$ = 0.446, explaining 44.6% of the variance). Being married or cohabiting was associated with higher odds of volunteering (adjusted odds ratio [AOR] = 1.66; 95% CI 1.07–2.57). Respondents holding at least a bachelor's degree were more than twice as likely to participate (AOR = 2.58; 95% CI 1.21–5.48). Employment identity showed strong effects: street- or community-level staff (AOR = 4.32; 95% CI 2.56–7.28) and healthcare staff (AOR = 2.48; 95% CI 1.31–4.67) volunteered more often than those in other occupations. Communist

**Table 3. Volunteer motivations (n = 674).**

| Motivation | Frequency | Response%[a] | Case%[b] |
|---|---|---|---|
| Helping people in need | 527 | 17.1% | 78.2% |
| Fulfilling social responsibility | 446 | 14.5% | 66.2% |
| Promoting community civility | 387 | 12.5% | 57.4% |
| Increasing social awareness | 358 | 11.6% | 53.1% |
| Developing personal skills | 309 | 10.0% | 45.8% |
| Influenced by volunteers or friends | 306 | 9.9% | 45.4% |
| Encouraged by family members or friends | 232 | 7.5% | 34.4% |
| Political party-membership requirement | 179 | 5.8% | 26.6% |
| Building social networks | 178 | 5.8% | 26.4% |
| Material or academic incentives | 158 | 5.1% | 23.4% |
| Other | 4 | 0.1% | 0.6% |
| Total | 3084 | 100.0% | 457.5% |

[a]Percentage of all responses (multiple selections allowed): Number of times a category was chosen ÷ Total number of choices made across all categories.

[b]Percentage of volunteers selecting the item: Number of volunteers who selected the category ÷ Total number of volunteers in the study.

**Table 4. Perceived barriers to volunteering (N = 1,023).**

| Barrier | Frequency | Response%[a] | Case%[b] |
|---|---|---|---|
| Lack of time | 618 | 27.0% | 60.3% |
| Insufficient professional skills | 507 | 22.2% | 49.5% |
| Fear of infection | 340 | 14.9% | 33.2% |
| Limited access to information | 337 | 14.7% | 32.9% |
| Inadequate protection of rights | 178 | 7.8% | 17.4% |
| Not interested | 114 | 5.0% | 11.1% |
| Lack of trust in organizations | 83 | 3.6% | 8.1% |
| Simply unwilling | 65 | 2.8% | 6.3% |
| Other | 44 | 1.9% | 4.3% |
| Total | 2286 | 100.0% | 223.1% |

[a]Percentage of all responses (multiple selections allowed): Number of times a category was chosen ÷ Total number of choices made across all categories.

[b]Percentage of volunteers selecting the item: Number of volunteers who selected the category ÷ Total number of volunteers in the study.

Party members also exhibited greater engagement (AOR = 1.68; 95% CI 1.07–2.64). Finally, each one-level increase on the five-point pre-2020 volunteering frequency scale tripled the likelihood of epidemic-period participation (AOR = 3.02; 95% CI 2.51–3.64). Sex, household registration, and age were not independently associated with volunteering after adjustment. All statistical tests were two-sided, and $p < 0.05$ was considered significant. After Bonferroni correction for multiple comparisons, the following variables remained statistically significant: occupation (street/community staff: AOR = 4.32, 95% CI: 2.56–7.28, $p < 0.001$; healthcare staff: AOR = 2.48, 95% CI: 1.31–4.67, $p < 0.05$) and frequency of volunteering before (AOR = 3.02, 95% CI: 2.51–3.64, $p < 0.001$) (S2 Table).

## Discussion

This current study set out to characterize the scale, patterns, and determinants of community volunteering during successive respiratory infectious-disease outbreaks in China. Three principal findings emerged from our analysis. First, resident engagement in epidemic-control activities was high: nearly two-thirds of respondents volunteered, and the median cumulative commitment reached 50 hours. Second, participation was driven chiefly by value-expressive motives, whereas time scarcity, perceived skill deficits, information gaps, and infection risk limited uptake. Third, higher educational attainment, being married or living with a partner, employment in community or health sectors, Communist Party membership, and volunteering experience before the COVID-19 pandemic independently facilitated engagement. The discussion below addresses each finding in turn, situates the results within the broader literature, and highlights practical implications.

The sheer volume of service from our findings exceeds pre- and early-pandemic benchmarks. The *Blue Book of Chinese Volunteering (2021–2022)* estimated that, by October 2021, the average volunteer hours per capita was about 7.44 nationwide [19]. In this present study, our participants reported a median of 50 hours over a comparable period. This divergence suggests that large-scale public-health contingencies may convert latent civic willingness into sustained commitments when community authorities provide visible, well-defined opportunities. Neighborhood committees could respond to staff shortages by adopting online and on-site recruitment, thereby expanding their volunteer base and creating a diverse repertoire of roles that ranged from queue management to supply delivery.

Motivational analysis confirmed the predominance of value-expression. Helping others, fulfilling social responsibility, and fostering community civility were each selected by more than half of volunteers, consonant with the six-function model

**Table 5. Multivariable logistic regression of factors associated with community epidemic-control volunteering (N = 1,023).**

| Variable | COR (95% CI) | AOR (95% CI) |
|---|---|---|
| Men | Reference | Reference |
| Women | 1.33 (1.03, 1.72)* | 0.87 (0.63, 1.21) |
| Age 18–34 | Reference | Reference |
| Age 35–44 | 1.45 (1.06, 1.98)* | 0.82 (0.50, 1.33) |
| Age 45–70 | 1.64 (1.19, 2.26)** | 1.26 (0.77, 2.06) |
| Urban | Reference | Reference |
| Rural | 0.78 (0.60, 1.01) | 0.79 (0.56, 1.12) |
| Married/living with partner | Reference | Reference |
| Single/divorced/widowed | 0.36 (0.27, 0.47)*** | 0.60 (0.39, 0.94)* |
| Primary/junior secondary | Reference | Reference |
| High-school/technical college/associate | 1.19 (0.65, 2.16) | 1.85 (0.87, 3.92) |
| Bachelor's degree or higher | 1.76 (1.00, 3.12) | 2.58 (1.21, 5.48)* |
| Employment: Other | Reference | Reference |
| Employment: Healthcare staff | 4.89 (2.80, 8.53)*** | 2.48 (1.31, 4.67)** |
| Employment: Student | 0.77 (0.52, 1.15) | 1.58 (0.78, 3.22) |
| Employment: Street/community staff | 8.71 (5.45, 13.93)*** | 4.32 (2.56, 7.28)*** |
| Political affiliation: unaffiliated | Reference | Reference |
| Communist Party member | 4.51 (3.11, 6.53)*** | 1.68 (1.07, 2.64) * |
| Democratic party member | 3.51 (0.41, 30.20) | 1.01 (0.10, 10.30) |
| Communist Youth League member | 0.70 (0.48, 1.03) | 0.55 (0.28, 1.07) |
| Frequency of volunteering before 2020 (5-level scale) | 3.521 (2.97, 4.17)*** | 3.02 (2.51, 3.64)*** |
| Total | — | — |

*p < 0.05; **p < 0.01; ***p < 0.001.

COR = crude odds ratio; AOR = adjusted odds ratio (mutually adjusted for all covariates).

of volunteer motivation [8]. Value expression, as conceptualized in the Volunteer Functions Inventory, refers to individuals' motivation to act on personally important values such as humanitarianism, altruism, and social responsibility [9,10]. In our context, this manifested as volunteers' desire to help others, fulfill social obligations, and contribute to community well-being during the health crisis. Previous work further shows that individuals with strong altruistic orientations and a desire for self-enhancement are especially likely to volunteer [9]. Because intrinsic motives dominate, non-pecuniary recognition and skill-oriented preparation are preferable to financial incentives. Field evaluations in China demonstrate that certificates, public commendations, posters, and awards strengthen responsibility, nurture a positive volunteer culture, and improve retention [13]. Brief, task-focused training can satisfy volunteers' self-development needs while ensuring competence for routine duties such as queue management and data entry.

Nevertheless, practical barriers continue to hamper recruitment and retention. Our study indicated that the five most frequently reported deterrents to epidemic-control volunteering were lack of discretionary time, insufficient professional skills, fear of infection, poor access to opportunity information, and inadequate protection of volunteer rights. Time scarcity is a well-established impediment; studies consistently show that limited free time markedly reduces the likelihood of service participation [5]. Perceived skill deficits, information gaps, and weak rights protection have likewise been identified as persistent shortcomings in the evolution of China's volunteer infrastructure [20,21]. Paradoxically, most epidemic-control roles in this study including queue management, data entry, supply distribution, require minimal technical expertise. The

mismatch suggests that volunteers' self-appraised incompetence reflects deficiencies in orientation and communication rather than genuine task complexity. Strengthening community-level training, standardizing role descriptions, and establishing an accessible information platform would therefore remove major entry barriers.

Although the Chinese Government has long encouraged citizen volunteering and has promulgated multiple protective regulations (e.g., the *Regulations on Voluntary Service*), material and organizational support at the community level remains insufficient, particularly for health-emergency roles [20,21]. Evidence shows that sound rights-protection mechanisms and competent organizational coordination are critical to the effectiveness as well as the sustainability of volunteer programs [22]. Accordingly, we recommend that local authorities formalize management structures, develop tiered training curricula, and implement clear legal safeguards so that residents can serve without apprehension.

Fear of infection constitutes a barrier unique to infectious disease contexts. Only 7.8 percent of volunteers in one national survey reported familiarity with relevant regulations, underscoring a knowledge deficit [23]. Even medical students, who possess comparatively strong infection-control literacy, cite the risk of contagion as a principal reason for declining service. According to a study from China, about half of undergraduates at a medical university selected "fear of infection" when asked why they would not volunteer for COVID-19 response [24]. International research echoed that fewer than one-third of medical students in a multinational survey were willing to serve during outbreaks of highly pathogenic infectious diseases, compared with 69% during natural disasters such as earthquakes [25]. To address these concerns, communities should combine basic infection-control training with tangible safeguards, including adequate personal protective equipment, online service options that reduce exposure, and comprehensive health-insurance coverage. Such measures are essential to reassure current volunteers and to cultivate a reliable reserve of community responders for future public-health emergencies.

Multivariable analysis identified five independent facilitators of epidemic-control volunteering: being married or cohabiting, holding a bachelor's degree or higher, working as street- or community-level staff or as healthcare personnel, Communist Party membership, and frequent pre-2020 volunteer activity. Each characteristic is consistent with theoretical and empirical work on social capital and civic engagement. Compared with residents with education at primary or junior secondary level, those with higher degrees generally enjoy more stable employment and broader social networks, place greater emphasis on non-material values, and therefore exhibit stronger willingness to undertake community service [6,11]. Studies from other settings further indicate that education and positive health behaviors buffer pandemic-related stigma [26] and thus may foster engagement in public-health initiatives. Marital status also mattered that individuals living with their spouse or partner volunteered more often than those who were single, divorced, or widowed. Stable family structures can provide logistical support, emotional encouragement, and normative reinforcement, all of which facilitate civic participation [16,17,27].

Occupational identity was another salient predictor. During periods of staff shortage, street-office employees and community workers were routinely redeployed to epidemic-control tasks, and many continued to serve in a voluntary capacity beyond their formal duties. Healthcare personnel likewise possessed relevant skills and professional obligation, which increased their probability of engagement. Political affiliation showed a similar pattern: Communist Party members were more likely to volunteer, plausibly reflecting the central government's call for party members to assume demonstrative roles in grassroots response [16]. Prior relevant experience also exerted a strong influence from our results. Residents who had volunteered frequently before 2020 were three times as likely to serve during COVID-19, a finding that accords with multiple studies linking past behavior to future civic action through habit formation and heightened role identity [6,28]. Familiarity with procedures and organizational contacts may also lower psychological barriers, making re-engagement more attractive.

The application of Bonferroni correction for multiple comparisons provided a more conservative assessment of these associations, with only the most robust predictors, occupation (street/community staff and healthcare personnel) and prior volunteer experience, maintaining statistical significance after correction. Variables such as education level, marital status,

and political affiliation, while demonstrating meaningful effect sizes in uncorrected analysis, did not survive the stringent correction threshold. These findings suggest that occupation and prior experience represent the most reliable predictors of volunteer participation, while other demographic factors may reflect borderline associations that warrant validation through larger-scale studies or alternative analytical approaches.

However, two variables, sex and age, often regarded as core predictors were not significant after adjustment in our analysis. One explanation is contextual that prolonged lockdowns confined many residents to their neighborhoods, equalizing discretionary time across demographic groups and diminishing traditional gender- or age-related differences. In routine settings, older adults may volunteer to offset social isolation after retirement [29,30], and homemakers often engage to compensate for their reduced workplace contacts [7]. Nevertheless, the widespread mobility restrictions during the outbreak provided ample free time for most residents, thereby potentially attenuating these usual patterns.

Collectively, these results have significant implications for emergency preparedness and community resilience systems in China and similar contexts. The robustness of occupation and prior experience as predictors suggests that emergency management agencies should prioritize targeted recruitment strategies focusing on healthcare workers, community staff, and individuals with established volunteer networks. Developing volunteer registries that maintain contact with experienced volunteers could facilitate rapid mobilization during future health emergencies. The 2024 revision of the Emergency Response Law establishes that social organizations and volunteers should participate in emergency response work under unified government command, though specific implementation guidelines are still under development [31]. Our findings on occupation and prior volunteer experience as key predictors can inform these implementation guidelines, particularly by providing evidence-based criteria for targeted recruitment and volunteer classification systems during health emergencies. The current proportion of residents (34%) who did not participate in volunteer activities, despite broad community support for volunteering, suggests important opportunities for expanding engagement. Our barrier analysis indicates that addressing time constraints, skill development needs, and infection control concerns could potentially convert non-participants into active volunteers, underscoring the importance of flexible volunteer opportunities and robust safety protocols to maximize community participation during health emergencies. Implementation strategies should also consider differences between urban and rural volunteer infrastructure, with urban areas benefiting from technology-enhanced platforms while rural communities may require emphasis on informal networks and locally adapted roles.

Several limitations exist in our current study and merit acknowledgment. First, the cross-sectional design and retrospective self-report introduce potential recall bias for activities spanning up to three years. Second, despite anonymity, social-desirability effects cannot be excluded. Third, our use of non-probability quota sampling via an online platform may have introduced selection bias, as internet access, digital literacy, and willingness to participate in online surveys can vary significantly across socioeconomic groups. This may also have resulted in a sample with higher educational attainment and digital engagement than the general population, potentially limiting the generalizability of our findings to all demographic groups. Fourth, unmeasured contextual variables, such as local outbreak severity or community governance quality, could confound observed associations. Fifth, our study was designed primarily to examine overall patterns of volunteer participation rather than detect regional differences, and the sample size of approximately 200 respondents per province may be insufficient for robust regional comparisons or interaction effect analyses. Therefore, future studies should incorporate longitudinal designs with larger, probability-based samples to validate borderline associations, utilize administrative service records to reduce recall bias, and apply mixed-methods approaches to better understand contextual factors influencing volunteer participation across diverse communities and emergency scenarios.

## Conclusion

Community volunteers constituted a substantial, intrinsically motivated workforce during recent respiratory epidemics in China. Based on the findings from our study, addressing time, skill, information, and safety barriers, while leveraging the enabling effects of education, family support, institutional affiliation, and prior experience, are critical to strengthen

community resilience and preparedness for future public-health emergencies. Implementing structured training, recognition, and legal-protection mechanisms would translate these insights into sustainable volunteer programs capable of rapid mobilization in the next crisis.

## Supporting information

**S1 Table. Multicollinearity assessment: Variance inflation factors for independent variables in logistic regression model.**
(S1_Table.PDF)

**S2 Table. Adjusted odds ratios for volunteer participation with multiple comparison correction using Bonferroni method.**
(S2_Table.PDF)

## Acknowledgments

The authors would like to thank all participants involved in this study.

## Author contributions

**Conceptualization:** Xinying Sun, Ying Ji.

**Data curation:** Huizi Jin.

**Formal analysis:** Huizi Jin.

**Funding acquisition:** Ying Ji, Fang Ning.

**Investigation:** Zhijing Li.

**Methodology:** Xinying Sun, Ying Ji.

**Project administration:** Xueting Ding, Xinying Sun, Ying Ji, Fang Ning.

**Supervision:** Ying Ji, Fang Ning.

**Validation:** Xueting Ding, Juan Cao.

**Visualization:** Huizi Jin.

**Writing – original draft:** Huizi Jin, Xueting Ding.

**Writing – review & editing:** Huizi Jin, Xueting Ding, Zhijing Li, Juan Cao, Ying Ji, Fang Ning.

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
