## [Decision Letter · Decision Letter 0]

5 Jul 2025

Dear Dr. Ji,

We look forward to receiving your revised manuscript.

Kind regards,

Vasuki Rajaguru, PhD

Academic Editor

PLOS ONE

Journal Requirements:

3. In the online submission form, you indicated that the datasets used during the current study are available from the corresponding author upon reasonable request.

Additional Editor Comments :

The Methods section requires revision to improve transparency and clarity in several key areas, including sampling limitations, participant recruitment flow, and variable validation.

Study Design: The use of a non-probability sampling method (quota sampling via an online platform) introduces potential selection bias. This is especially important given that internet access, digital literacy, and willingness to participate in online surveys can vary significantly across different socioeconomic groups. This limitation should be explicitly acknowledged in the manuscript's limitations section. Participants and Sampling: The sample size of approximately 200 respondents per province may be insufficient to detect regional differences or interaction effects unless justified with a power calculation or sensitivity analysis. The authors should clarify whether statistical weights were applied to better align the sample with national demographic distributions. Furthermore, the rationale for selecting only five provinces should be provided, along with an explanation for excluding other populous or epidemiologically important provinces such as Sichuan and Shanghai. Statistical Analysis: The analysis section lacks details on whether multidisciplinary among independent variables was assessed prior to performing multivariable logistic regression. Additionally, the manuscript does not describe how missing data were handled—whether through list-wise deletion, imputation, or another method. The authors should also clarify how the significance level was adjusted, if at all, to account for multiple comparisons. Overall, the statistical analysis section should be revised to include these methodological details and ensure that appropriate statistical procedures were applied. Also recommended to include the significance and policy implications of the study.

Reviewers' comments:

Reviewer's Responses to Questions

**Comments to the Author**

1. Is the manuscript technically sound, and do the data support the conclusions?

Reviewer #1: Yes

Reviewer #2: Yes

2. Has the statistical analysis been performed appropriately and rigorously?

Reviewer #1: Yes

Reviewer #2: Yes

3. Have the authors made all data underlying the findings in their manuscript fully available?

Reviewer #1: Yes

Reviewer #2: Yes

4. Is the manuscript presented in an intelligible fashion and written in standard English?

Reviewer #1: Yes

Reviewer #2: Yes

Reviewer #1: General Comments:

This is a valuable and timely study that contributes to understanding community-based epidemic response mechanisms. The high volunteer participation rates and identification of enabling and inhibiting factors offer useful guidance for public health policy and emergency planning.

Reviewer #2: This manuscript presents a timely and well-structured cross-sectional study exploring the prevalence and determinants of community volunteering during respiratory infectious disease outbreaks in China. The topic is highly relevant, the methodology is sound, and the findings offer practical implications for public health preparedness. The paper is largely well-written, though certain sections would benefit from improved clarity, better theoretical integration, and expanded contextualization.

Please see the attachment for additional details.

**Do you want your identity to be public for this peer review?** For information about this choice, including consent withdrawal, please see our Privacy Policy

Reviewer #1: **Yes: ** Dr Suraj Abdulkarim Abdullahi

Reviewer #2: No

---

## [Author Response · Author response to Decision Letter 1]

22 Jul 2025

Dear Dr. Rajaguru and reviewers,

Thank you so much for your thorough review and constructive feedback on our manuscript. We greatly appreciate the opportunity to strengthen our work through your insightful comments. We have carefully addressed each point raised and have made corresponding revisions to improve the manuscript, particularly the transparency and clarity of our Methods and Discussion sections. Below, we provide detailed responses to each comment below (and "Response to Reviewers.doc" for better formatting). All line numbers referenced correspond to the clean version of the revised manuscript:

Response to Editor Comments

Editorial Requirements

Response: Thank you very much for providing the style templates. We have thoroughly updated our manuscript to comply with PLOS ONE's formatting requirements, including proper file naming conventions, author and affiliation formatting, subsection titles, reference styling, and body text formatting according to the provided templates.

Response: We really appreciate your guidance on the data sharing requirements. We have uploaded our minimal de-identified dataset to the figshare data repository and updated our Data Availability statement accordingly.

The statement now reads: “The minimal de-identified dataset used in the current study is available from [https://figshare.com/articles/dataset/Data_for_the_study_Community_volunteer_participation_and_its_determinants_during_respiratory_infectious_disease_outbreaks_in_China_a_cross-sectional_study_across_multiple_provinces_/29602721?file=56387153].”

3. In the online submission form, you indicated that the datasets used during the current study are available from the corresponding author upon reasonable request.

Response: As noted in the response to comment#2, we made our de-identified dataset publicly available through the figshare repository to meet PLOS ONE’s data availability requirements. The dataset is now accessible to other researchers for verification and further analysis.

Additional Editor Comments

4. The Methods section requires revision to improve transparency and clarity in several key areas, including sampling limitations, participant recruitment flow, and variable validation.

Study Design: The use of a non-probability sampling method (quota sampling via an online platform) introduces potential selection bias. This is especially important given that internet access, digital literacy, and willingness to participate in online surveys can vary significantly across different socioeconomic groups. This limitation should be explicitly acknowledged in the manuscript's limitations section.

Response: We completely agree with this important feedback. Given the post-pandemic context and associated constraints, an online survey approach was selected to enhance sample representativeness and ensure timely data collection. We have now explicitly acknowledged this limitation in our Discussion section under limitations.

We now have: “… our use of non-probability quota sampling via an online platform may have introduced selection bias, as internet access, digital literacy, and willingness to participate in online surveys can vary significantly across socioeconomic groups. This may also have resulted in a sample with higher educational attainment and digital engagement than the general population, potentially limiting the generalizability of our findings to all demographic groups” (lines 383-387).

5. Participants and Sampling: The sample size of approximately 200 respondents per province may be insufficient to detect regional differences or interaction effects unless justified with a power calculation or sensitivity analysis.

Response: Thank you for this valuable point. We acknowledge that our study was primarily designed to examine overall patterns of volunteer participation rather than detect regional differences. The target sample size was determined using the formula N = [Z²1-α/2P(1-P)]/d², where α = 0.05, Z1-α/2 = 1.96, and the acceptable margin of error d = 0.10. Based on previous research estimating 30% prevalence of relevant outcomes during the COVID-19 pandemic, a minimum of 81 participants per province was required. Our achieved sample of approximately 200 respondents per province exceeded this threshold and was adequate for our primary objective of identifying factors associated with volunteer participation overall.

We have clarified this in our revised Methods section by adding: “We used the formula N = [Z²1-α/2P(1-P)]/d², where α = 0.05, Z1-α/2 = 1.96, the acceptable margin of error d = 0.10, and the 30% prevalence of relevant outcomes during the COVID-19 pandemic to estimate the target sample size. As a result, we got a minimum required sample of 81 participants per province [16]” (lines 114-117) and “…leaving 1,023 valid cases (effective response rate 99.8 %). This satisfied our sample size calculation requirement of 81 participants per province (405 in total)” (lines 135-136). We also acknowledged the limitation regarding regional comparisons in the Discussion: “Fifth, our study was designed primarily to examine overall patterns of volunteer participation rather than detect regional differences, and the sample size of approximately 200 respondents per province may be insufficient for robust regional comparisons or interaction effect analyses” (lines 389-392).

6. The authors should clarify whether statistical weights were applied to better align the sample with national demographic distributions.

Response: We did not apply statistical weights in our analysis. Our quota sampling approach was designed to approximate national distributions for key demographic variables (sex, age, and household registration).

We updated out Methods-Statistical analysis section to clarify this: “No statistical weights were applied to the data, as our quota sampling approach was designed to approximate national demographic distributions” (lines 191-192).

In our Methods- Recruitment and sampling section, we had described: “We used quota sampling to approximate the national population distribution reported in the Seventh National Census in China. Quotas were set for sex (men 51 %, women 49 %), age (18–39 years 40 %, 40–70 years 60 %), and Chinese household registration (urban 65 %, rural 35 %)” (lines 120-122) to demonstrate that our sampling design aligned with national demographic distributions to ensure that our sample composition closely mirrors the target population characteristics.

7. Furthermore, the rationale for selecting only five provinces should be provided, along with an explanation for excluding other populous or epidemiologically important provinces such as Sichuan and Shanghai.

Response: Thank you for requesting this clarification. We have now provided a detailed rationale for province selection in our Methods section. The five provinces were selected based on multiple criteria: (1) regions with the highest cumulative COVID-19 cases from each of the five major geographical regions (east, west, south, north, and central China), (2) economic development diversity, and (3) geographical distribution to represent different regions of mainland China.

We added: “The selection of survey provinces considered multiple criteria including selecting areas with the highest cumulative COVID-19 case counts before November 2022 from each of the five major geographical regions (northern, southern, eastern/central, western, and north‑eastern), economic development diversity, and geographical distribution to ensure representative coverage of mainland China [16]” to our Method section (lines 110-114).

8. Statistical Analysis: The analysis section lacks details on whether multidisciplinary among independent variables was assessed prior to performing multivariable logistic regression.

Response: Thank you for this important methodological question. We conducted comprehensive multicollinearity assessment using variance inflation factors (VIF) prior to performing our multivariable logistic regression analysis. All VIF values ranged from 1.032 to 1.270, well below the commonly accepted threshold of 5, indicating acceptable levels of multicollinearity among independent variables.

We have added this information to our Statistical Analysis section: “Prior to conducting multivariable logistic regression, we assessed multicollinearity among independent variables using variance inflation factors (VIFs). All VIF values ranged from 1.032 to 1.270 (S1 Table 1), well below the threshold of 5, indicating acceptable levels of multicollinearity” (lines 187-190).

We also added Supplementary 1 Table 1 to show the exact numbers of VIFs.

9. Additionally, the manuscript does not describe how missing data were handled—whether through list-wise deletion, imputation, or another method.

Response: Thanks a lot for requesting clarification on this important aspect of our methodology. Our study design eliminated missing data through mandatory response requirements implemented via the Wenjuanxing platform. We set the platform's system configuration to require participants completing all questionnaire items before submission was permitted, resulting in complete data for all study variables without the need for imputation or deletion methods.

We have clarified this in our Data Collection section of the Methods: “We set the online questionnaire platform to require mandatory completion of all questions, with participants unable to submit responses without answering every item, thereby ensuring complete data collection for all study variables” (lines 136-139).

10. The authors should also clarify how the significance level was adjusted, if at all, to account for multiple comparisons. Overall, the statistical analysis section should be revised to include these methodological details and ensure that appropriate statistical procedures were applied.

Response: We really appreciate this comment. We improved our analysis by adding Bonferroni correction for multiple comparisons in our logistic regression. After correction, only the most robust predictors maintained statistical significance: occupation (street/community staff: AOR = 4.315, 95% CI: 2.559-7.276, p<0.001; healthcare staff: AOR = 2.476, 95% CI: 1.313-4.670, p<0.05) and frequency of volunteering before 2020 (AOR = 3.021, 95% CI: 2.513-3.636, p<0.001). Variables such as education, marital status, and political affiliation, while showing meaningful effect sizes in uncorrected analysis, did not survive the conservative correction threshold.

We have made the following additions to address this:

• Methods-Statistical analysis section: “To account for multiple comparisons, Bonferroni correction was applied with appropriate adjustment of significance levels” (lines 192-194).

• Results section: “After Bonferroni correction for multiple comparisons, the following variables remained statistically significant: occupation (street/community staff: AOR = 4.32, 95% CI: 2.56-7.28, p<0.001; healthcare staff: AOR = 2.48, 95% CI: 1.31-4.67, p<0.05) and frequency of volunteering before (AOR = 3.02, 95% CI: 2.51-3.64, p<0.001) (S1 Table 2)” (lines 239-43). We also added Supplementary 1 Table 2 to show the exact numbers.

• Discussion section: “The application of Bonferroni correction for multiple comparisons provided a more conservative assessment of these associations, with only the most robust predictors, occupation (street/community staff and healthcare personnel) and prior volunteer experience, maintaining statistical significance after correction. Variables such as education level, marital status, and political affiliation, while demonstrating meaningful effect sizes in uncorrected analysis, did not survive the stringent correction threshold. These findings suggest that occupation and prior experience represent the most reliable predictors of volunteer participation, while other demographic factors may reflect borderline associations that warrant validation through larger-scale studies or alternative analytical approaches” (lines 342-350).

11. Also recommended to include the significance and policy implications of the study.

Response: Thank you very much for this recommendation. We have expanded our Discussion section to better highlight the policy significance and practical implications of our findings in lines 359-379:

“Collectively, these results have significant implications for emergency preparedness and community resilience systems in China and similar contexts. The robustness of occupation and prior experience as predictors suggests that emergency management agencies should prioritize targeted recruitment strategies focusing on healthcare workers, community staff, and individuals with established volunteer networks. Developing volunteer registries that maintain contact with experienced volunteers could facilitate rapid mobilization during future health emergencies. The 2024 revision of the Emergency Response Law establishes that social organizations and volunteers should participate in emergency response work under unified government command, though specific implementation guidelines are still under development [31]. Our findings on occupation and prior volunteer experience as key predictors can inform these implementation guidelines, particularly by providing evidence-based criteria for targeted recruitment and volunteer classification systems during health emergencies. The current proportion of resident

---

## [Decision Letter · Decision Letter 1]

7 Aug 2025

Community volunteer participation and its determinants during respiratory infectious disease outbreaks in China: a cross-sectional study across multiple provinces

PONE-D-25-28388R1

Dear Dr. Ying Ji,

We’re pleased to inform you that your manuscript has been judged scientifically suitable for publication and will be formally accepted for publication once it meets all outstanding technical requirements.

Within one week, you’ll receive an e-mail detailing the required amendments. When these have been addressed, you’ll receive a formal acceptance letter, and your manuscript will be scheduled for publication.

Kind regards,

Vasuki Rajaguru, PhD

Academic Editor

PLOS ONE

Additional Editor Comments (optional):

All the required comments are amended. 

Reviewers' comments:

Reviewer's Responses to Questions

**Comments to the Author**

Reviewer #1: All comments have been addressed

Reviewer #2: All comments have been addressed

2. Is the manuscript technically sound, and do the data support the conclusions?

Reviewer #1: Yes

Reviewer #2: Yes

3. Has the statistical analysis been performed appropriately and rigorously?

Reviewer #1: Yes

Reviewer #2: Yes

4. Have the authors made all data underlying the findings in their manuscript fully available?

Reviewer #1: Yes

Reviewer #2: Yes

5. Is the manuscript presented in an intelligible fashion and written in standard English?

Reviewer #1: Yes

Reviewer #2: Yes

Reviewer #1: Additional Comment:

The manuscript presents a comprehensive and well-structured study that addresses a relevant and timely research question. The methodology is sound, and the findings contribute valuable insights to the field. The authors have demonstrated a strong understanding of the subject matter, and the data analysis appears rigorous. However, I recommend the authors clarify certain aspects of the discussion section to better contextualize their findings within existing literature. Additionally, minor grammatical and typographical errors should be addressed to improve the manuscript's overall readability.

Recommendation:

I recommend the manuscript for acceptance pending minor revisions as outlined above. The study's significance and contribution to the field justify its publication, and I believe that with these adjustments, it will meet the journal’s standards for quality and clarity.

Reviewer #2: The revisions have thoroughly addressed my previous comments, and the manuscript has improved significantly in clarity and overall quality. Methodological details have been well clarified, and the expanded discussion effectively highlights the policy implications. I consider the manuscript suitable for publication in its current form.

**Do you want your identity to be public for this peer review?** For information about this choice, including consent withdrawal, please see our Privacy Policy

Reviewer #1: **Yes: ** DR SURAJ ABDULKARIM

Reviewer #2: No

---

## [Editor Report · Acceptance letter]

PONE-D-25-28388R1

PLOS ONE

Dear Dr. Ji,

I'm pleased to inform you that your manuscript has been deemed suitable for publication in PLOS ONE. Congratulations! Your manuscript is now being handed over to our production team.

Kind regards,

on behalf of

Dr. Vasuki Rajaguru

Academic Editor

PLOS ONE